# Spin-orbit torque manipulation of sub-terahertz magnons in antiferromagnetic $\alpha$-Fe$_2$O$_3$

Dongsheng Yang[1,3], Taeheon Kim[1,2,3], Kyusup Lee[1], Chang Xu ⬡[1], Yakun Liu[1], Fei Wang[1], Shishun Zhao[1], Dushyant Kumar ⬡[1] & Hyunsoo Yang ⬡[1] ✉

The ability to electrically manipulate antiferromagnetic magnons, essential for extending the operating speed of spintronic devices into the terahertz regime, remains a major challenge. This is because antiferromagnetic magnetism is challenging to perturb using traditional methods such as magnetic fields. Recent developments in spin-orbit torques have opened a possibility of accessing antiferromagnetic magnetic order parameters and controlling terahertz magnons, which has not been experimentally realised yet. Here, we demonstrate the electrical manipulation of sub-terahertz magnons in the $\alpha$-Fe$_2$O$_3$/Pt antiferromagnetic heterostructure. By applying the spin-orbit torques in the heterostructure, we can modify the magnon dispersion and decrease the magnon frequency in $\alpha$-Fe$_2$O$_3$, as detected by time-resolved magneto-optical techniques. We have found that optimal tuning occurs when the Néel vector is perpendicular to the injected spin polarisation. Our results represent a significant step towards the development of electrically tunable terahertz spintronic devices.

Magnons, the quanta of collective spin-wave excitations, carry the spin angular momentum without moving charges[1,2], and could serve as potential information carriers, allowing Joule-heat-free data transfer, These features have made them of great interest in the energy-efficient information technology[3–5]. Antiferromagnets, where two-spin sublattices are aligned antiparallel due to the negative exchange interaction, host antiferromagnetic (AFM) magnons which have frequencies that are 2–3 orders of magnitude higher than their ferromagnetic counterparts due to a stronger AFM exchange energy[6,7], and insulating antiferromagnets have low magnetic dissipation[8]. These features have rendered insulating antiferromagnets as a promising platform for energy-efficient terahertz (THz) magnonics and optoelectronics.

Such devices require active electronic control of THz magnons. One such approach is to utilise current-induced spin-orbit torques (SOT)[9,10]. However, attention has principally been confined to manipulating the static magnetic state, leading to AFM magnetic

reorientation/switching[11–14]. In stark contrast, the direct control of the THz spin dynamics in antiferromagnets, such as the magnon frequency, is critical for novel applications such as THz spin nano-oscillators[15–17] and rectifiers[18–21]. Pioneering efforts to control the magnon frequency have focused on conventional ferromagnetic and ferrimagnetic thin films and two-dimensional exfoliated flakes, whose magnon frequencies reside in tens of GHz[22–28]. Although theoretical works[15,17,29,30] pointed out that the frequency of the uniform resonant mode (wavevector $k = 0$ magnon) in antiferromagnets can be controlled by SOT, the experimental demonstration, including propagating magnons ($k \neq 0$), has remained elusive.

Here, we demonstrate the electrical control of the sub-THz magnon frequency in an easy-plane AFM insulator, $\alpha$-Fe$_2$O$_3$. For this work, we use 5-mm-thick, (0001)-oriented $\alpha$-Fe$_2$O$_3$ crystals (Fig. 1a, b). Figure 1c, d show the X-ray diffraction pattern and atomic force microscopy image of the $\alpha$-Fe$_2$O$_3$ sample, respectively, confirming the high

[1]Department of Electrical and Computer Engineering, National University of Singapore, Singapore, Singapore. [2]Electro-Medical Device Research Centre, Korea Electrotechnology Research Institute, Ansan, Republic of Korea. [3]These authors contributed equally: Dongsheng Yang, Taeheon Kim.
✉e-mail: eleyang@nus.edu.sg

crystal quality and the [0001] crystallographic orientation. The magnetisation versus temperature is displayed in Fig. 2a. The magnetic phase transition in $\alpha$-Fe$_2$O$_3$, called the Morin transition, occurs at the

Morin temperature $T_M$, which is ~261 K. The magnetic configuration above $T_M$ is schematically illustrated in Fig. 1b. The magnetic order of two sublattices, $\mathbf{m_1}$ and $\mathbf{m_2}$, is nearly antiparallel but slightly canted due

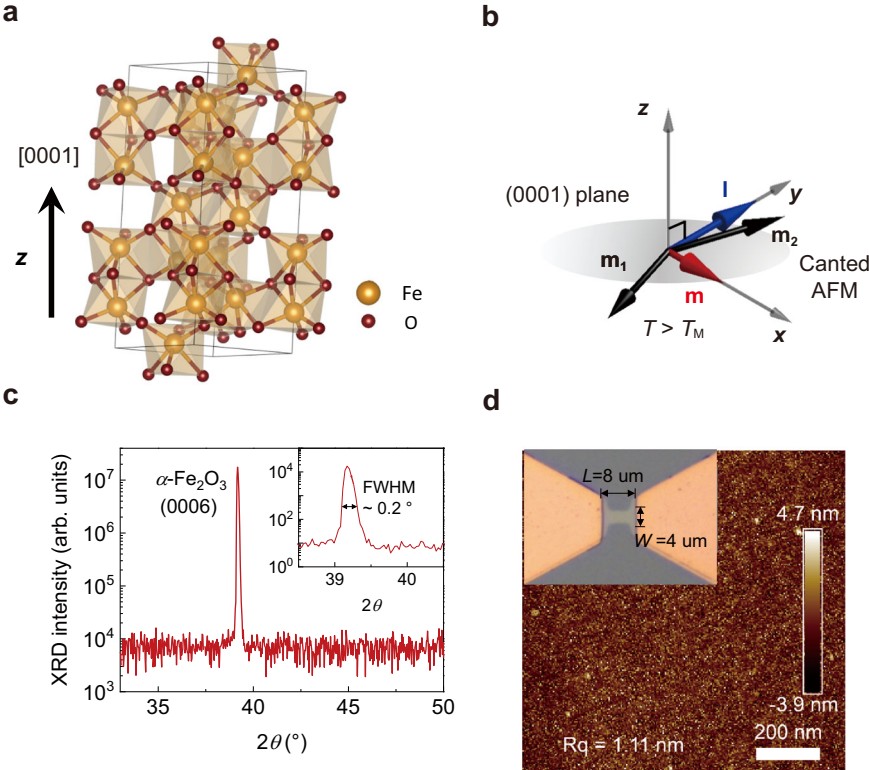

**Fig. 1 | Material characterisation of $\alpha$-Fe$_2$O$_3$ sample. a, b** Crystal (**a**) and magnetic (**b**) structure of $\alpha$-Fe$_2$O$_3$ above the Morin temperature $T_M$ = 261 K. **m, l, m$_1$** and **m$_2$** represent a canted moment, Néel vector, and 1st and 2nd sublattice moment. **c** X-ray diffraction (XRD) measurement of the $\alpha$-Fe$_2$O$_3$ sample. The peak at 39.2° matches well with the position of the $\alpha$-Fe$_2$O$_3$ (0006) peak. Inset: the zoomed view

of the $\alpha$-Fe$_2$O$_3$ (0006) peak showing the full width at half maximum (FWHM). **d** Atomic force microscopy image of the $\alpha$-Fe$_2$O$_3$ (0001) sample. The root mean square average (Rq) of the profile heights is 1.11 nm. Inset: the device image with a dimension of 4 μm in width (W) and 8 μm in length (L).

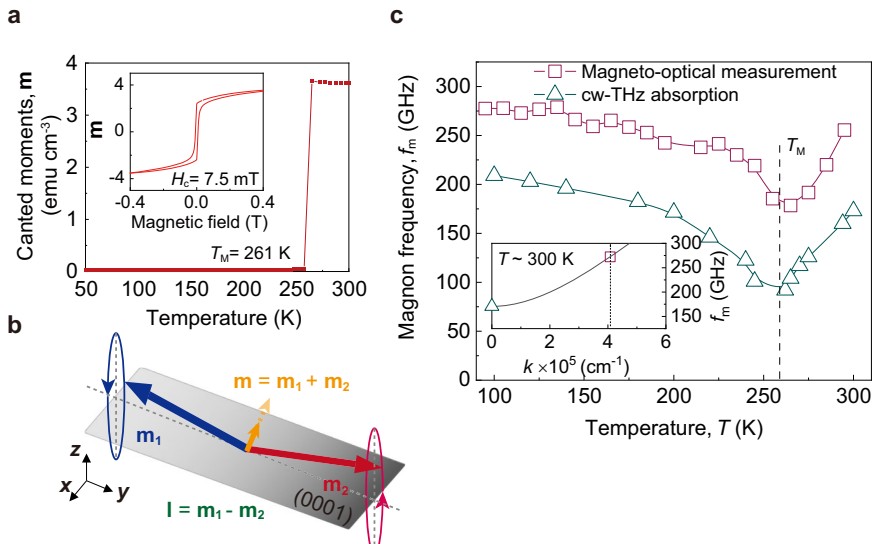

**Fig. 2 | Temperature dependence of $\alpha$-Fe$_2$O$_3$ magnon frequency $f_m$ without applying currents. a** Canted moments **m** as a function of temperature from 50 and 300 K, showing a Morin transition at $T_M$ ~ 261 K. Inset: the magnetic hysteresis loop with an in-plane coercivity $H_c$ = 7.5 mT at T = 300 K. **b** Configuration of q-AFM spin resonance. Both **l** and **m** are located in the (0001) plane and perpendicular to each other. **c** Temperature dependences of $f_m$ (wavevector $k > 0$) obtained by magneto-

optical measurement (squares) and $f_m$ ($k = 0$) by continuous-wave terahertz (cw-THz) absorption measurement (triangles). The overall frequency blueshift measured by magneto-optical measurements is evidence of finite $k$ magnons. Inset: the calculated $\alpha$-Fe$_2$O$_3$ magnon dispersion relation for the q-AFM mode along the [0001] direction, in which both $f_m$ and $k$ match well with two measurement results.

to the Dzyaloshinskii-Moriya interaction (DMI)[31], resulting in a nearly unit-length Néel vector $\mathbf{l} = (\mathbf{m}_2 - \mathbf{m}_1)/2$ and a perpendicular canted magnetic moment $\mathbf{m} = (\mathbf{m}_1 + \mathbf{m}_2)/2$ and on the (0001) plane. The small coercivity around 7.5 mT (the inset of Fig. 2a) results from the insignificant in-plane magneto-anisotropy of $\alpha$-Fe$_2$O$_3$. As a result, a weak external magnetic field (**H**) could align the direction of **m**, leading to a magnetic configuration with $\mathbf{l} \perp \mathbf{H}$ and $\mathbf{m} // \mathbf{H}$ in the (0001) plane. At room temperature, two eigenmodes of magnon in $\alpha$-Fe$_2$O$_3$ correspond to the out-of-plane and in-plane elliptical precessional motion of $\mathbf{m}_1$ and $\mathbf{m}_2$ lying in the (0001) plane[32]. Our study focuses on the out-of-plane precessional mode, also known as the quasi-antiferromagnetic (q-AFM) mode, which exhibits sub-THz magnon frequencies (Fig. 2b)[33]. In contrast, the frequency of the in-plane mode, or the quasi-ferromagnetic (q-FM) mode, resides in the range of several GHz in the absence of **H**[34,35].

## Results

### Sub-THz magnon measurements in $\alpha$-Fe$_2$O$_3$ with currents

In this work, we use the time-resolved magneto-optical method for observing the sub-THz magnons in $\alpha$-Fe$_2$O$_3$ (Methods, Supplementary Note 1). This method is a benchmark technique to investigate the spin dynamics in AFM insulators through magneto-optical interactions, i.e., the Faraday effect and Cotton-Mouton effect. Specifically, the spin dynamics in $\alpha$-Fe$_2$O$_3$ are obtained via ultrafast excitation of the pump pulse with the photon energy of 3.1 eV above the $\alpha$-Fe$_2$O$_3$ bandgap and detected by tracking the polarisation rotation of the probe pulse using typical magneto-optical detection, similar to previous experiments[36–39] (Supplementary Notes 2 and 3). We adopt this method as it is sensitive to the interfacial effect where SOT tuning prevails and can measure higher-frequency magnon components with finite $k$[26,40].

To identify the value of magnon wavevector $k$ measured by the time-resolved magneto-optical method, we first measure the resonance ($k = 0$ magnon) frequency by using a continuous-wave THz (cw-THz) absorption method (Supplementary Note 4)[41]. The temperature-dependent q-AFM magnon frequency $f_m$ of $\alpha$-Fe$_2$O$_3$ without the current using both methods are summarised in Fig. 2c. The results of cw-THz absorption method indicate a resonance frequency of approximately 200 GHz at room temperature with a dip in the vicinity of the $T_M$, which is consistent with the q-AFM resonance mode ($k = 0$ magnon) in $\alpha$-Fe$_2$O$_3$[33]. In comparison, the $f_m$ measured by the time-resolved magneto-optical method displays a similar trend but is consistently shifted by ~70 GHz over that by the cw-THz absorption method. This consistent blueshift in frequency constitutes concrete evidence for the existence of finite $k$ magnons, similar to the previous report on AFM DyFeO$_3$[40]. In this case, the value of probed $k$ can be determined using the Bragg equation[42] $k = 2k_0 n(\lambda_0)\cos\delta'$ in which $k_0$ is the wavenumber of the probe light, $n(\lambda_0)$ is the refractive index of $\alpha$-Fe$_2$O$_3$/Pt heterostructure at the centre wavelength of the probe beam $\lambda_0 = 800$ nm and $\delta'$ is the refraction angle of the probe beam at an incidence angle $\delta$. By applying our experimental parameters[43] with $n(\lambda_0)$ equating 2.6 at $\lambda_0 = 800$ nm and $\delta$ being 0°, the measured $k$ in the time-resolved magneto-optical experiment is estimated to be $4.08 \times 10^5$ cm$^{-1}$ (Supplementary Note 5).

To quantitatively correlate the $f_m$ with different $k$, the magnon dispersion of $\alpha$-Fe$_2$O$_3$ along [0001] or the $z$ direction (Supplementary Notes 6 and 7) is given by

$$f_m(kJ_c) = \sqrt{(v_0 k)^2 + f_{m,k=0}^2(J_c)}, \qquad (1)$$

where $v_0 \sim 32$ km s$^{-1}$ is the limiting magnon velocity along the [0001] crystalline direction[44], $f_{m,k=0}$ is the q-AFM magnon frequency at $k = 0$, and $J_c$ is the applied charge current. The value of $f_m$ at $k = 4.08 \times 10^5$ cm$^{-1}$ is larger than that of $f_{m,k=0}$ as it is shifted by $v_0 k$. Here, $\lambda_0$ is dominant in determining $k$, whereas the $\delta$ dependence on $k$ is small; $\delta = 0°$ ($\delta = 30°$) corresponds to $k = 4.08 \times 10^5$ cm$^{-1}$ ($k = 4 \times 10^5$ cm$^{-1}$) and $f_m = 272$ GHz ($f_m = 269$ GHz). As shown in the inset of Fig. 2c, both $f_m$

and $k$ extracted from both time-resolved magneto-optical and cw-THz absorption methods agree well with the calculated magnon dispersion, as a cross-check of the $k$ value extracted by the Bragg equation. In addition, we observe a slow oscillation mode at ~55 GHz (Fig. 3). This corresponds to a laser-induced acoustic phonon mode, which has been widely studied previously[40,42] (Supplementary Note 8).

### SOT manipulation of sub-THz $\alpha$-Fe$_2$O$_3$ magnons

Next, we explore the SOT manipulation for the $\alpha$-Fe$_2$O$_3$ magnons at $k = 4.08 \times 10^5$ cm$^{-1}$. As illustrated in Fig. 3a, the adjacent Pt is a strong spin-orbit coupling material and acts as the spin source. By applying $J_c$ across the Pt layer, transverse spin accumulations are generated at the $\alpha$-Fe$_2$O$_3$/Pt interface with the spin polarisation $\sigma$. The spin accumulation induces a SOT on the magnetic order of $\alpha$-Fe$_2$O$_3$ and generates magnon current density $J_m$, allowing for the manipulating of $f_m$. To understand the modified magnon dispersion, we adopt a magnon-mediated spin current $J_m$, which can propagate into $\alpha$-Fe$_2$O$_3$ (Method and Supplementary Note 9). By monitoring the polarisation rotation of the reflected probe pulses under different $J_c$, the manipulation of the $f_m$ can be studied as a function of $J_c$ (Fig. 3b, c).

To confirm the dominant mechanism of SOT rather than other unwanted effects (e.g., heating and strain), we perform the measurements under both $\sigma // \mathbf{m}$ and $\sigma // \mathbf{l}$ configurations. Due to the large size of the magnetic domain in the $\alpha$-Fe$_2$O$_3$ sample, which exceeds hundreds of micrometres, our device can be considered in a saturated mono-domain condition (Methods and Supplementary Fig. 13). Figure 3c, d summarises the $f_m$ with different $J_c$ and under both configurations ($\sigma // \mathbf{l}$ and $\sigma // \mathbf{m}$). The direction of **m** in $\alpha$-Fe$_2$O$_3$ is aligned along the external magnetic field **H** and is orthogonal to the orientation of **l** in the (0001) plane, as confirmed by spin Hall magnetoresistance measurements (Supplementary Fig. 14). No magnetic field is applied during the time-resolved magneto-optical measurement.

At the $\sigma // \mathbf{m}$ configuration, we observe that $f_m$ decreases nonlinearly with increasing $J_c$ from 272 GHz at $J_c = 0$ A cm$^{-2}$ to 258 GHz at $J_c = 2 \times 10^7$ A cm$^{-2}$, resulting in a $f_m$ tuning of $-14$ GHz (the minus sign refers to a redshift of $f_m$). We note that this redshift of $f_m$ is opposite to the heating-induced effect that increases $f_m$ when $T > T_M$. Moreover, we find that the tuning effect is symmetrical with the polarity of $J_c$, which is distinct from that of ferromagnets[26]. This is reasonable as $J_c$ does not break the symmetry of **l** in $\alpha$-Fe$_2$O$_3$[8]. Furthermore, we find that the frequency of the acoustic phonon $f_p$ remains unchanged with $J_c$, which supports that the tuning of $f_m$ is due to spin-related mechanisms rather than the strain resulting from the current flow. We further developed a comprehensive spin-wave model that combined the SOT and AFM magnon dispersion to explain our experimental observations (Supplementary Notes 7 and 8). In the $\sigma // \mathbf{m}$ configuration, SOT reorients **l** from the equilibrium axis (the $y$-axis in Fig. 3d), reducing the effective anisotropy and finally resulting in the redshift of $f_m$. As shown in Fig. 3c, the simulation result in a solid line matches well with both the polarity dependence and the amount of $f_m$ change.

We then study the second configuration with $\sigma // \mathbf{l}$. As shown in Fig. 3c, we observe a similar redshifted trend in $f_m$ by increasing the current density, with a reduced amount of $-6$ GHz tuning at $J_c = 2 \times 10^7$ A cm$^{-2}$, ~43% of that at $\sigma // \mathbf{m}$. Therefore, $f_m$ reveals distinct dependences on the configuration, showing that the direction of **l** with respect to $\sigma$ is important for efficient $f_m$ tuning. This anisotropic feature is analogous to the previous report on the $\alpha$-Fe$_2$O$_3$ spin transport, where spins propagate longer with $\sigma // \mathbf{l}$ than at $\sigma // \mathbf{m}$[8]. Our result indicates that SOT is less pronounced at $\sigma // \mathbf{l}$ than at $\sigma // \mathbf{m}$, leading to a reduced $f_m$ tuning. Furthermore, the simulated $f_m$ at $\sigma // \mathbf{l}$ in Fig. 3c matches the experimental results.

## Discussion

To further confirm the role of SOT on the change in $f_m$, we conduct a control measurement in the $\alpha$-Fe$_2$O$_3$/Cu (5 nm) device, where we have

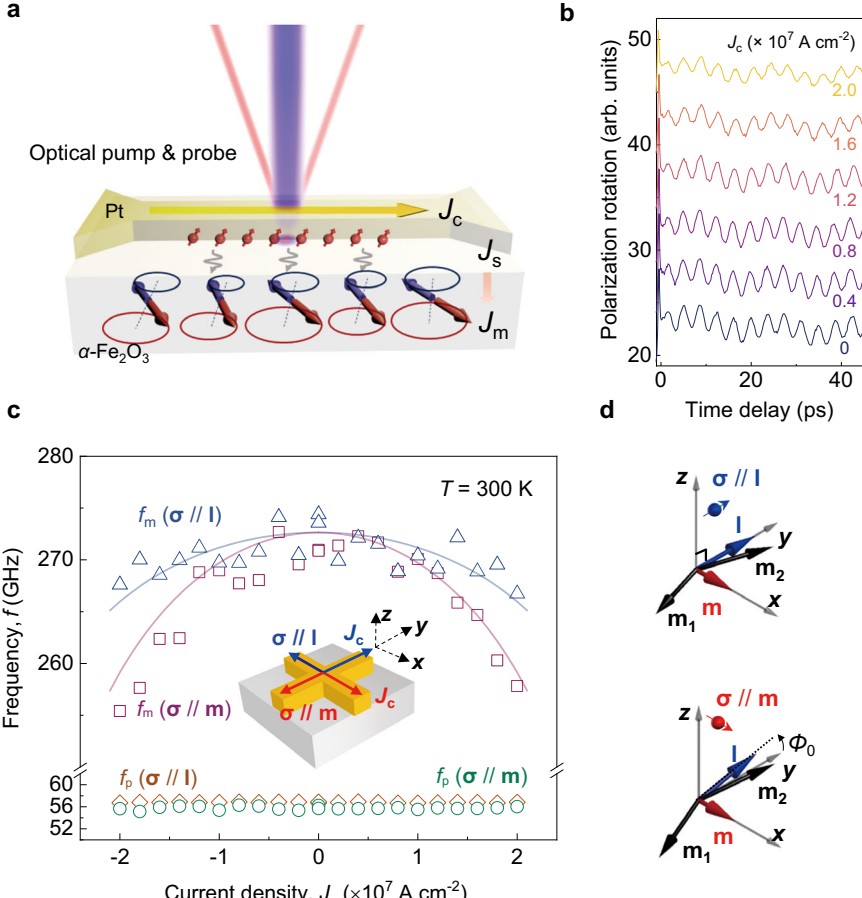

**Fig. 3 | Experimental observation of SOT-manipulated q-AFM $f_m$ of $\alpha$-Fe$_2$O$_3$ at $T$ = 300 K. a** Schematics of the SOT-tuned AFM magnon measurement using the time-resolved magneto-optical method (Methods). Coherent magnons in $\alpha$-Fe$_2$O$_3$ are excited via femtosecond optical pumping and manifest the polarisation rotation of the probe beam via magneto-optical interaction. $J_c$, $J_s$, and $J_m$ are the injected charge current, spin current and magnon current, respectively. **b** $f_m$ is manipulated by $J_s$ injected from the adjacent Pt layer and detected as the oscillation signal using a balanced detection scheme ($\sigma$ // **m** configuration). $\sigma$: spin polarisation. The overlapped phonon oscillation at phonon frequency $f_p$ = 55 GHz can be used to judge SOT tuning. **c** Summary of SOT-tuned $f_m$ and $f_p$ with $J_c$ at both $\sigma$ // **l** and $\sigma$ // **m** configurations. $f_m$ at $\sigma$ // **l** ($\sigma$ // **m**) is denoted by triangles (squares) and $f_p$ at $\sigma$ // **l** ($\sigma$ // **m**) is denoted by diamonds (circles). The solid lines represent the theoretical calculation results (Supplementary Note 7). **d** SOT-manipulated spin configurations at $\sigma$ // **l** and $\sigma$ // **m**, where $\Phi_0$ is SOT-induced reorientation of **l**.

used Cu, due to its negligible spin-orbit coupling strength. No change is observed in $f_m$ at either $\sigma$ // **m** or $\sigma$ // **l** configuration (Fig. 4a). These results confirm the SOT origin of our observed $f_m$ tuning in Fig. 3c and rule out other strain and heating effects. Next, we consider the current-induced heating effect. The limited temperature rise may slightly increase $f_m$ at $T > T_M$ (Fig. 2c) but cannot account for the observed decrease of $f_m$ in Fig. 3c (Supplementary Note 10). Therefore, we conclude that SOT dominates the magnon tuning in our experiments. Figure 4b summarises the tunability and $f_m$ in various magnetic materials by electrical means (Supplementary Table 1 for details). The SOT effect on the spin lifetime in Supplementary Note 11 shows no change for both spin configurations as the applied current density is smaller than that for self-oscillation.

Noticeably, tuning $f_m$ using SOT is significantly more energy efficient compared to current-induced Oersted fields, $H_{Oe}$. The induced $H_{Oe}$, by Ampère's law ($H_{Oe} \sim \mu_0 J_C t_{Pt}/2$) is as small as $H_{oe}$ = 0.63 mT for $J_c$ = $2 \times 10^7$ A cm$^{-2}$, where $\mu_0$ is the vacuum permeability and $t_{Pt}$ is the thickness of Pt layer. The $f_m$ change induced by $H_{Oe}$ is expected to be $\gamma H_{Oe}/2\pi \approx 0.017$ GHz, where $\gamma$ is the electron gyromagnetic ratio, which is nearly $10^3$ times smaller in magnitude than the manipulation of $f_m$ achieved by SOT.

We calculate the magnon dispersion relation at $\sigma$ // **m** as a function of $J_c$ and plot in Fig. 4c (Supplementary Notes 6 and 7). As $J_c$ increases, the dispersion relation curves near the $k$ = 0 are redshifted

downward. However, the amount of frequency shift at large $k$ is smaller. Therefore, maximised frequency tuning occurs for the resonant mode ($k$ = 0), resulting in a frequency tunability of $-24$ GHz/ ($2 \times 10^7$ A cm$^{-2}$). The SOT increases the band curvature and thus increases the magnon group velocity, $v_g = 2\pi \times \partial f_m/\partial k$. As shown in Fig. 4d, for $k = 4.08 \times 10^5$ cm$^{-1}$ magnons, $v_g$ increases from 25.3 to 26.7 km s$^{-1}$ at $J_c$ = $2 \times 10^7$ A cm$^{-2}$ with $\sigma$ // **m**. Furthermore, we explore the SOT-induced $f_m$ modulation with varying the exchange field $H_E$ and effective magnetic anisotropy field $H_{eff,K}$ for $\sigma$ // **m** (Supplementary Note 12). In this case, antiferromagnets with a high $H_E$ and low $H_{eff,K}$ are preferred for efficient $f_m$ tuning, which is in line with the theoretical result that IrMn with a much higher $H_{eff,K}$ ~ 5000 Oe yields a reduced tuning efficiency[45].

As shown in Fig. 4b, our work marks the electrically manipulated $f_m$ at the highest frequency (270 GHz for $k = 4.08 \times 10^5$ cm$^{-1}$ and 170 GHz for $k$ = 0) ever reported. For magnetic materials with various values of $f_m$, it is fair to compare their tuning efficiency, defined as $\eta(J_c) = [f_m(J_c) - f_m(0)]/[J_c \times f_m(J_c)]$, under a normalised input amplitude. The efficiency of our $\alpha$-Fe$_2$O$_3$/Pt heterostructure $\eta_{Fe2O3} = -6.8\%/$ ($10^7$ A cm$^{-2}$) is more than five times higher than the benchmark ferrimagnetic yttrium iron garnet (YIG) with $\eta_{YIG} \sim -1.1\%/(10^7$ A cm$^{-2}$)[22]. The magnetic field dependence of $f_m$ in the same $\alpha$-Fe$_2$O$_3$ sample is also measured. No clear change in $f_m$ is observed up to 0.2 T magnetic field (Supplementary Note 13). This further supports the effectiveness of

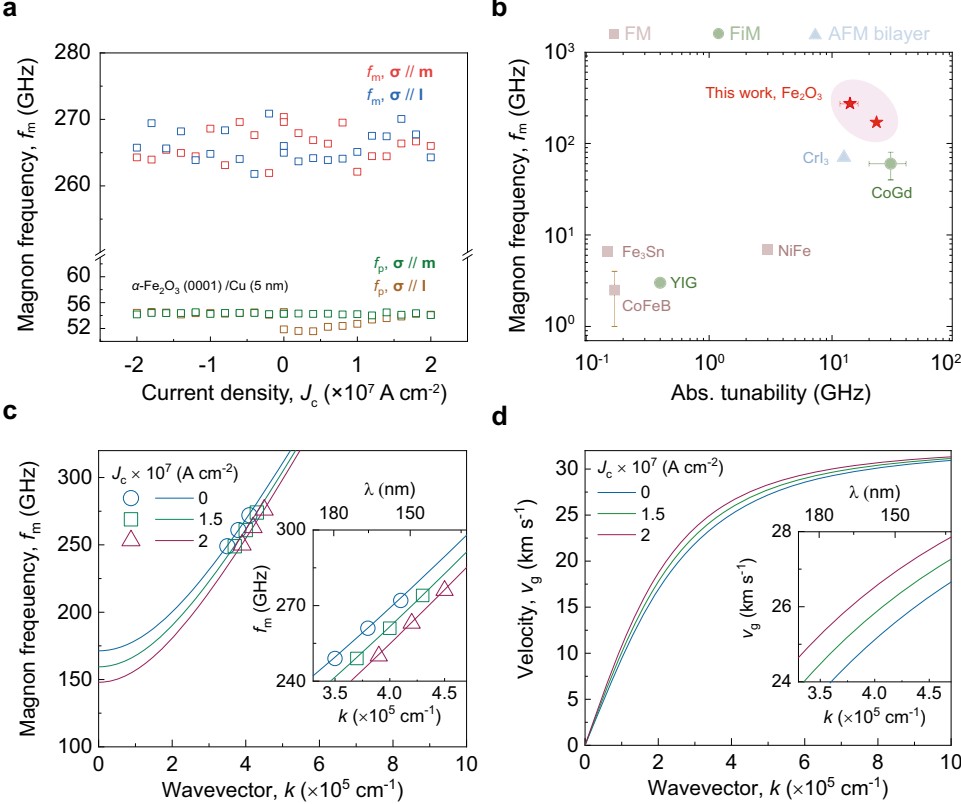

**Fig. 4 | Control measurement of $\alpha$-Fe$_2$O$_3$/Cu (5 nm) and simulation results of SOT-tuned $\alpha$-Fe$_2$O$_3$ magnon dispersion. a** Current-dependence of $f_m$ and $f_p$ in the controlled $\alpha$-Fe$_2$O$_3$ (0001)/Cu (5 nm) device under two-spin configurations. Neither $f_m$ nor $f_p$ is tuned by $J_c$ at both spin configurations, which suggests that SOT is essential for the observed $f_m$ tuning in Fig. 3. **b** Comparison of the tunability and $f_m$ by electrical means summarised in Supplementary Table 1. Two red stars represent $f_m$ our tuning results for $k = 0$ and $k = 4.08 \times 10^5$ cm$^{-1}$. The error bars indicate the uncertainty of the data extracted from references. FM: ferromagnet, FiM: Ferrimagnet, AFM: antiferromagnet. **c** Numerical (symbols) and analytical (lines) calculated magnon dispersion in the $\sigma$ // $m$ configuration. Maximum $f_m$ tunability of $-23$ GHz occurs for the zone centre (resonant) mode at $J_c = 2 \times 10^7$ A cm$^{-2}$. **d** Magnon group velocity with $k$ extracted from **c**. The magnon group velocity increases effectively for intermediate $k$, leading to an increase from 25.3 to 26.7 km s$^{-1}$ at $J_c = 2 \times 10^7$ A cm$^{-2}$ for $k = 4.08 \times 10^5$ cm$^{-1}$.

the SOT-tuning in $\alpha$-Fe$_2$O$_3$ magnon. Although other stimuli such as strain[46], Joule heating[23,47] and magnetic fields[3] can potentially realise a similar or even greater tunability, the application scenarios are severely limited by their substantial energy consumption and poor device scalability. SOT-tuning requires only a nanometre-thick spin source, which is a critical feature for the miniaturisation of future spintronic devices.

Our work provides an efficient and universal means to manipulate sub-THz magnons electrically in antiferromagnets. The SOT manipulation of magnons is applicable to a wide range of AFM materials without utilising exotic spin textures or complicated device structures. It, therefore, can play a significant role in developing all-electrical THz magnonic and photonic devices. For example, it can be used to create on-chip integrated THz photonics nano-circuits[48].

## Methods

### (0001) cut $\alpha$-Fe$_2$O$_3$ single crystal

The $\alpha$-Fe$_2$O$_3$ single-crystalline sample with a thickness of 1 mm was purchased from MaTecK Material-Technologie & Kristalle GmbH (Germany). The sample surface was polished with a surface roughness of about 1 nm and a miscut orientation angle below 0.1°.

### Device fabrication

Prior to device fabrication, the $\alpha$-Fe$_2$O$_3$ sample surface was cleaned with acetone, isopropyl alcohol (IPA) and deionised (DI) water by a standard ultrasonic bath process. A maskless ultraviolet lithography system (TTT-07-UVlitho, TuoTuo Technology (Singapore) Pte. Ltd) was used for device patterning with channel dimensions of 4 μm wide

and 8 μm long for subsequent deposition (inset of Fig. 1d). The Pt layer was deposited on $\alpha$-Fe$_2$O$_3$ with a thickness of 5 nm using DC magnetron sputtering at a base pressure of $9 \times 10^{-9}$ Torr. A 4 nm-thick SiO$_2$ insulating layer was deposited to protect the device from oxidation.

### Spin Hall magnetoresistance measurements

In response to an external field, the orientations of magnetic order in $\alpha$-Fe$_2$O$_3$ are coupled with spin currents injected from the adjacent Pt layer via the spin Hall effect, manifesting the spin Hall magnetoresistance (SMR)[49]. The AC longitudinal harmonic signals for Hall bar devices were measured in a physical property measurement system (PPMS, Quantum Design) at room temperature. During the measurement, a constant amplitude sinusoidal current with a frequency of 13.7 Hz was applied to the devices by a Keithley 6221 current source. The angle-dependent first-harmonic longitudinal voltage $V_f$ was recorded by a Stanford research SR830 lock-in amplifier to obtain the first-harmonic longitudinal resistance $R_f$ while rotating the in-plane magnetic field of 1 T. The negative sign of SMR with a value of $\Delta R/R \sim 0.2$ % indicates an excellent interfacial quality of spin injection into $\alpha$-Fe$_2$O$_3$ (Supplementary Note 14).

### Time-resolved magneto-optical measurements

In the time-resolved magneto-optical setup, the probe beam was the output of an 80 MHz Ti: sapphire femtosecond laser system (Spectra-Physics Mai Tai, pulse duration of 75 fs) centred at 1.55 eV (800 nm), and the pump beam was the second harmonic by using a $\beta$-Barium borate single crystal to 3.1 eV (400 nm). Both pump and probe pulses were linearly polarised. The fluence ratio between the pump and probe

beams was around 10:1. Both beams were focused onto the sample surface at normal incidence by a ×50 objective lens (the pump beam diameter is ~1.8 μm with a typical fluence of 1.0 mJ cm$^{-2}$; the probe beam diameter is ~1.2 μm with a typical fluence of 100 μJ cm$^{-2}$). The magnon detection depth is the order of the penetration length of probe beams in $\alpha$-Fe$_2$O$_3$, which is around 300 nm[50]. In this case, the reduced amplitude of SOT away from the $\alpha$-Fe$_2$O$_3$/Pt interface potentially leads to the broadening of linewidth and reduced amplitude of magnon spectra in the magneto-optical results (Supplementary Note 15). A continuous flow cryostat system (ST-300, Janis) was used to set the sample temperature between 77 and 300 K with fluctuations below 0.1 K. For the applied current, square-wave-liked current pulses (a pulse amplitude: 0-4 mA; a pulse width: 20 μs; duty cycle: 2%) were generated by using a current source (Keithley 6221) and synchronised with the femtosecond laser beams through a function generator (DS345, Stanford Research). Reflectivity and the polarisation rotation of the probe beams were measured by using a Wollaston prism and a balanced detector combination (Supplementary Fig. 1). To be noted, the pure magnetic linear birefringence cannot be demodulated by using an HWP. However, in cases where both magnetic linear birefringence and dichroism simultaneously manifest, both the polarisation and ellipticity of the probe beam become subject to tuning. In such circumstances, the oscillating signal emanating from magnons can be detected through the utilisation of either an HWP or by strategically manipulating the optical axis of a Wollaston prism.

In order to excite $\alpha$-Fe$_2$O$_3$ magnons efficiently, a 400 nm (3.1 eV) femtosecond pulse is used for pumping. The optical radiation with photon energy larger than the bandgap (2.14 eV)[50] of $\alpha$-Fe$_2$O$_3$ induces the strong electronic transition from O$^{2-}$ to Fe$^{3+}$ ions[51] (Supplementary Note 2). The spin excitation, mediated by the intense electronic transition, is strongly confined near the surface due to a finite penetration depth (~29 nm) of the pump light. Therefore, the spatially non-uniform and transient excitation ignites broadband magnons[40]. The oscillation signal of the manipulated magnons is detected by the polarisation rotation of the probe beam via the magnetic-optical interactions. It occurs due to the different refractive indices of the probe between the one parallel to the **l** and the other perpendicular to it on the $\alpha$-Fe$_2$O$_3$ (0001) plane, which shows a sinusoidal signal as a function of the polarisation angle of probe light (Supplementary Fig. 4b). The limitation of the measurement scheme is that the thermal effect disturbs the magnon signal at higher $J_c > 2 \times 10^7$ A cm$^{-2}$, which prevents from conducting any meaningful analysis (details are shown in Supplementary Note 16). Therefore, we restrict the data analysis to $J_c < 2 \times 10^7$ A cm$^{-2}$ in this work.

### Magnetic domain measurement

At room temperature, we measure the magnetic domain of the $\alpha$-Fe$_2$O$_3$(0001) single crystal using magneto-optical Kerr microscopy. Longitudinal geometry is employed to probe the canted in-plane magnetic moment of $\alpha$-Fe$_2$O$_3$(0001). Similar to the earlier report[8], we observe distinct magnetic domains with a domain size over hundreds of micrometres (Supplementary Note 17). As this domain size much exceeds the size of SOT-tuned devices, it is reasonable to consider the main results under mono-domain conditions.

### Atomistic spin simulations for $\alpha$-Fe$_2$O$_3$ magnon

We construct an atomistic Landau-Lifshitz-Gilbert equation, including the damping-like SOT

$$\partial \mathbf{m}_i / \partial t = -\gamma \mathbf{m}_i \times \mathbf{H}_{\text{eff},i} + \beta \mathbf{m}_i \times \partial \mathbf{m}_i / \partial t + J_{\text{m},z} \mathbf{m}_i \times (\mathbf{m}_i \times \boldsymbol{\sigma}), \quad (2)$$

where $\mathbf{m}_i$ is the unit vector of magnetisation at site $i$, $\mathbf{H}_{\text{eff},i}$ is the effective field, and $\boldsymbol{\sigma}$ is the spin polarisation. We use the following parameters of $\alpha$-Fe$_2$O$_3$ at room temperature: the exchange field $H_E = 930$ T, the hard-axis anisotropy field $H_{Kz} = -16.8$ mT, the basal anisotropy field $H_B = 1$ μT, which originates from spontaneous

magnetostriction[52], the Dzyaloshinskii-Moriya field $H_D = 2.5$ T, and $\gamma = 1.76 \times 10^{11}$ T s$^{-1}$. $J_{\text{m},z}$ is the magnon spin current injected from Pt (Supplementary Note 9). To verify the dispersion relation, we apply an external ac magnetic field $H_{\text{ext}} = 10$ mT × sinc($2\pi f t$) and spatially distribute the magnetic field $H_{\text{ext}} = 0.1$ mT × sinc($k(z-z_0)$) for the ignition of broadband magnons where $z_0$ is the surface of $\alpha$-Fe$_2$O$_3$ and magnons propagate along the $z$ direction. The DC magnon spin current is applied to the entire AFM system with the polarisation along the $x$ and $y$ directions. Numerical simulations were conducted from 0 to 1 μs with a minimum time interval of 0.1 ps with 10,000 spins. Magnon spin current is applied over a distance from 0 to 150 nm. The same parameters are used in theoretical and numerical calculations.

## Data availability

All other data that support the plots within this paper and other findings of this study are available from the Supplementary Note or the corresponding author upon reasonable request.

## Code availability

The code that has been used for this work is available from the corresponding author upon reasonable request.

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

## Acknowledgements

The research was supported by the National Research Foundation (NRF) Singapore Investigatorship (NRFIO6-2020-0015).

## Author contributions

Y.L. and D.Y. fabricated the devices. D.Y., F.W. and D.K. performed material characterisations. D.Y. and K.L. performed the time-resolved experiments. D.Y., K.L. and T.K. analysed the data. T.K. performed the theoretical and numerical calculations. C.X. conducted simulations for current-induced heating. S.Z. conducted static MOKE measurements. H.Y. conceived the experiments and supervised the project. All authors contributed to the writing of the manuscript.

## Competing interests

The authors declare no competing interests.
