## [Peer Review File · Nature Communications]

Reviewers' Comments:

Reviewer #1:

Remarks to the Author:

Review to the manuscript "Spin-orbit torque manipulation of sub-terahertz magnons in antiferromagnetic α -Fe₂O₃"

The manuscript titled "Spin-orbit torque manipulation of sub-terahertz magnons in antiferromagnetic α -Fe₂O₃" by Dongsheng Yang, Taeheon Kim, Kyusup Lee, Chang Xu, Yakun Liu, Fei Wang, Shishun Zhao, Dushyant Kumar, and Hyunsoo Yang is recommended for publication after a major revision. The manuscript explores the electrical manipulation of sub-terahertz magnons in the α -Fe₂O₃/Pt antiferromagnetic heterostructure. The authors demonstrate that applying spin-orbit torques in the heterostructure can modify the magnon dispersion and reduce the magnon frequency in α -Fe₂O₃, as observed through time-resolved magneto-optical techniques. They have identified the optimal tuning condition when the Néel vector is perpendicular to the injected spin-polarized current. The work is timely, addressing a topic that has gained prominence in various fields of physics over the last decade and recently attracted significant attention in spintronics. It not only reveals interesting phenomena from a general perspective but also holds great potential for future spintronic devices by leveraging complex coupled dynamics. Considering the timely results and detailed analysis, I believe that Nature Communications is an appropriate journal for this manuscript.

However, I suggest a further revision of the manuscript to address some minor points:

- The authors mention an anisotropy field of 0.1 mT in the easy plane in their theoretical model. It is important to clarify the physical nature of this field. As it is well known (see e.g. Morrish (1994) in "Canted antiferromagnetism: hematite") hematite exhibits rhombohedral symmetry (group D_{3d}6). The quasi-ferromagnetic gap at zero field is primarily determined by spontaneous magnetostriction rather than the easy-plane anisotropy, which is extremely small (approximately 10⁻⁶ T). It is crucial to provide references to the relevant literature to support this point (e.g., refer to the Morrish's book).
- It is necessary to provide numerical estimates for the critical current density at which self-oscillations can occur in the heterostructures. This information will help readers understand the experimental findings more comprehensively.
- To enhance the interest of a broader readership, it is advisable to include references to potential applications of the experimental and theoretical research data. For example, references such as [Safin, A, et al. 2022. Magnetochemistry 2022, Vol. 8, No. 26; A. Mitrofanova, et al. Appl. Phys. Lett. 2022. Vol. 120, 072402; G. Consolo, et al. Physical Review B. 2021. Vol. 103. No. 134431; P. Popov, et al. Phys. Rev. Appl. 2020. Vol. 13, 044080] can be cited to demonstrate the practical relevance of the findings.

Reviewer #2:

Remarks to the Author:

The paper by Yang et al. presents an experimental study of electrical frequency control of sub-terahertz magnons in an antiferromagnet using interfacial spin-orbit torque (SOT). The authors demonstrate a decrease in the frequency by about 7% when an electric current is passed through the Pt layer deposited on top of a bulk Fe₂O₃ crystal. The observed effects can be useful for controlling future terahertz spintronic devices. Of course, it would be more exciting if the authors demonstrated electrical tuning of the magnon lifetime, as was done earlier for ferromagnets. Nevertheless, the demonstration of electrical frequency tuning can also be seen as an important step forward in the field of antiferromagnetic spintronics.

I am not completely sure of the correctness of the authors' interpretation of the experimental data. I will be glad, if the authors can convince me that my concerns are groundless.

1. Taking into account the fast decay of the spin current with increasing distance from the interface, the SOT effects are limited to a nanometer-thick layer of Fe₂O₃ near the interface. I find it difficult to understand how one can speak about (modified) magnon dispersion, if the properties of the medium are changed on a spatial scale much smaller than the wavelength of magnons. It is particularly unclear when magnons propagating perpendicular to the interface are considered. This needs to be clarified.
2. At what depth are magnons detected by the magneto-optical technique? As far as I understand,

to achieve efficient Bragg reflection, this depth must be at least several wavelengths of magnons, i.e. several hundred nanometers. At such depths, the effects of SOT become negligible (see the comment above). Thus, it is unclear, how to correctly interpret the results of the measurements. I would expect the detected magnons to have dispersion unaffected by SOT.

3. From a comparison of the data shown in Fig. 2c and the data in Fig. 3c ($J_c=0$), it can be seen that the deposition of Pt leads to an increase in the frequency of magnons from about 250 to 270 GHz. In other words, the presence of Pt strongly modifies the properties of the system. Therefore, it is inaccurate to refer to the data obtained for bare Fe₂O₃ when discussing SOT effects in Fe₂O₃/Pt. In particular, in Fe₂O₃/Pt, the temperature dependence can differ from that shown in Fig. 2c. The temperature dependence must be measured for the Fe₂O₃/Pt heterostructure. This should not be a problem for the magneto-optical technique.

4. How can we be sure that the observed effects are due to SOT in Fe₂O₃ and not due, for example, to modification of proximity magnetism in Pt?

5. Why are measurements of the main effect (Fig. 3c) limited by $J_c=2 \times 10^7$ A/cm²? According to the authors' calculations, the increase of the temperature is rather small at these densities. Thus, the measurement range can be easily extended to much higher currents. It should be possible to go up to 10^8 , as was done in other works for similar dimensions of Pt electrodes. If the authors are correct in their interpretation, then the frequency should start to increase at a certain current due to the effects of temperature. These data must be added to make the interpretation convincing.

6. An optional but very interesting extension would be a measurement similar to that shown in Fig. 3c, with the laser spot placed on the surface of bare Fe₂O₃ next to the Pt electrode. These data could very convincingly demonstrate that the effects exist only underneath the Pt strip.

Reviewer #3:

Remarks to the Author:

Yang et al. reported spin-orbit torque manipulation of sub-terahertz magnons in antiferromagnet. Authors detect sub-terahertz spin precession in alpha-Fe₂O₃ by time-resolved pump-probe technique with injecting current in the heterostructure. They observed first time that spin precession frequency of antiferromagnetic alpha-Fe₂O₃ can be modulated by the spin-orbit torque in alpha-Fe₂O₃/Pt heterostructure. The polarity of precession frequency change is opposite with the trend of Joule heating; therefore, the change is certainly caused by the spin-orbit torque. I feel that this study contains many novelties therefore this study is valuable to be published in Nature Communications. But I have following questions and comments, which must be addressed before acceptance.

1. Authors used second-order magneto-optical effect such as Cotton-Mouton effect to detect antiferromagnetic resonance. Usually, this effect is birefringence effect and cannot be detected by measuring rotation of electric field. Actually, the previous study (Ref. 29) used quarter-wave plate to detect ellipticity of probe beam, however, authors use half-wave plate to detect rotation of probe beam. It should be mentioned that why they measured rotation instead of ellipticity in the revised manuscript. Authors mentioned in Supplementary Information section 3 that q-FM magnon mode reverses with the direction of magnetic field, but they only compare with and without magnetic field in Extended Data Fig. 5.

2. In Fig. 3c, it looks that experimental data has large scattering, i.e., there is deviation between experimental symbols and solid curves. Where does this scattering come from? In Fig. 4b and Extended Data Table 1, authors defined tuning efficiency of frequency and compare with other reports. Is it better to show error bar in the tuning efficiency because there is scattering in the experimental data?

3. In Fig. 4a, authors show the results of control samples, that is alpha-Fe₂O₃/Cu. But the range of vertical axis is different with the case for alpha-Fe₂O₃/Pt sample (Fig. 3c). It is better to compare with same range of vertical axis.

4. As shown mainly in Supplementary Information, the authors explain the antiferromagnetic resonance by the free energy including homogeneous and inhomogeneous exchange energy, uniaxial magnetic anisotropy energy, and DM interaction energy, but I cannot find the parameters (values) they used exactly in the manuscript. This must be described in the revised version.

5. Authors derived equations for antiferromagnetic resonance frequency in Supplementary information but the details of parameters are not written in the text. For example, resonance

frequency is expressed in terms of effective magnetic field due to exchange interaction or DM interaction, but their correspondence with energy (i.e., a , A , D) are not described. This must be added for readers.

6. As shown in Fig. 3c, d, the precession frequency is quadratically decreased with increasing current for both cases that spin is parallel and perpendicular to Neel vectors. This trend is explained by the theory of antiferromagnetic magnon including spin-orbit torque term as shown in Eq. (S12a) and Eq. (S13a), where the spin-orbit torque modifies $k = 0$ magnon frequency as well as equilibrium magnetization angle. By analyzing the results, authors can obtain the parameters of spin-orbit torque effective magnetic field or spin-Hall angle in Pt, however, they did not describe exact values in the text. This must be addressed in the revised version.

7. Regarding spin-orbit torque strength parameter ζ , this strength is inversely proportional to thickness of antiferromagnetic layer, but the thickness of the antiferromagnetic layer is 1 nm as described in Methods section, quite thin, so the spin-orbit torque strength becomes quite weak. In the Atomistic spin simulation section, authors state that the effective thickness of antiferromagnetic layer used was 4 nm. It must be stated why they chose this value.

Reviewer #1:

Review to the manuscript “Spin-orbit torque manipulation of sub-terahertz magnons in antiferromagnetic α -Fe₂O₃”. The manuscript titled “Spin-orbit torque manipulation of sub-terahertz magnons in antiferromagnetic α -Fe₂O₃” by Dongsheng Yang, Taeheon Kim, Kyusup Lee, Chang Xu, Yakun Liu, Fei Wang, Shishun Zhao, Dushyant Kumar, and Hyunsoo Yang is recommended for publication after a major revision. The manuscript explores the electrical manipulation of sub-terahertz magnons in the α -Fe₂O₃/Pt antiferromagnetic heterostructure. The authors demonstrate that applying spin-orbit torques in the heterostructure can modify the magnon dispersion and reduce the magnon frequency in α -Fe₂O₃, as observed through time-resolved magneto-optical techniques. They have identified the optimal tuning condition when the Néel vector is perpendicular to the injected spin-polarised current. The work is timely, addressing a topic that has gained prominence in various fields of physics over the last decade and recently attracted significant attention in spintronics. It not only reveals interesting phenomena from a general perspective but also holds great potential for future spintronic devices by leveraging complex coupled dynamics. Considering the timely results and detailed analysis, I believe that Nature Communications is an appropriate journal for this manuscript. However, I suggest a further revision of the manuscript to address some minor points:

Our reply: We sincerely express our gratitude to the reviewer for his/her invaluable suggestions and comments, which have played a pivotal role in refining our research. In the revised manuscript and the response provided below, we have carefully addressed each of the reviewer's comments in a point-by-point manner.

1. The authors mention an anisotropy field of 0.1 mT in the easy plane in their theoretical model. It is important to clarify the physical nature of this field. As it is well known (see e.g. Morrish (1994) in "Canted antiferromagnetism: hematite") hematite exhibits rhombohedral symmetry (group D_{3d}6). The quasi-ferromagnetic gap at zero field is primarily determined by spontaneous magnetostriction rather than the easy-plane anisotropy, which is extremely small (approximately 10⁻⁶ T). It is crucial to provide references to the relevant literature to support this point (e.g., refer to the Morrish's book).

Our reply: We sincerely thank the reviewer's comments. We agree with the reviewer and we modified the origin of easy plane anisotropy as follows in Method.

“...the basal anisotropy field $H_B = 1 \mu\text{T}$, which originates from spontaneous magnetostriction [A. H. Morrish, Canted antiferromagnetism: hematite, World Scientific, 1994.], ...”

2. It is necessary to provide numerical estimates for the critical current density at which self-oscillations can occur in the heterostructures. This information will help readers understand the experimental findings more comprehensively.

Our reply: As shown in the Extended Data Fig. 9a, after $J_c = 3 \times 10^7 \text{ A cm}^{-2}$, the Neel vector reorientation by SOT is not stable (f_m drops to 0) because the SOT reduces effective anisotropy at J_c . Therefore, $J = 3 \times 10^7 \text{ A cm}^{-2}$ is defined as the critical current density J_{cr} for self-oscillation. In the modified manuscript, we added the numerical estimation of J_{cr} to indicate self-oscillation in Supplementary Section 12. The discussion has been added in the Supplementary Section 12 as follows:

“...The theoretical limit for the maximum frequency change by SOT is worth noting. For $\sigma // \mathbf{m}$, the SOT compensates the effective anisotropy at $J_c^{\text{th}, \perp \sigma} \approx 3 \times 10^7 \text{ A cm}^{-2}$ (Extended Data Fig. 9), above which the right-angle oscillation around another axis occurs...”

3. To enhance the interest of a broader readership, it is advisable to include references to potential applications of experimental and theoretical research data. For example, references such as [Safin, A, et al. 2022. Magnetochemistry 2022, Vol. 8, No. 26; A. Mitrofanova, et al. Appl. Phys. Lett. 2022. Vol. 120, 072402; G. Consolo, et al. Physical Review B. 2021. Vol. 103. No. 134431; P. Popov, et al. Phys. Rev. Appl. 2020. Vol. 13, 044080] can be cited to demonstrate the practical relevance of the findings.

Our reply: Thanks for this helpful comment. We have added these papers in the modified manuscript as references 20, 21, 29, and 30.

Reviewer #2:

The paper by Yang et al. presents an experimental study of electrical frequency control of sub-terahertz magnons in an antiferromagnet using interfacial spin-orbit torque (SOT). The authors demonstrate a decrease in the frequency by about 7% when an electric current is passed through the Pt layer deposited on top of a bulk Fe₂O₃ crystal. The observed effects can be useful for controlling future terahertz spintronic devices. Of course, it would be more exciting if the authors demonstrated the electrical tuning of the magnon lifetime, as was done earlier for ferromagnets. Nevertheless, the demonstration of electrical frequency tuning can also be seen as an important step forward in the field of antiferromagnetic spintronics.

I am not completely sure of the correctness of the authors' interpretation of the experimental data. I will be glad if the authors can convince me that my concerns are groundless.

Our reply: We would like to thank the reviewer for finding our work useful and commenting with helpful inputs to improve our work. Please see our point-by-point responses below.

We included the SOT effect on the spin lifetime in the main text and Supplementary Section 11 as shown in Figure R1. The trend is consistent with our theoretical model that the spin lifetime is not changed before reaching self-oscillation. We added descriptions on page 7 in the manuscript as follows:

“Besides, the SOT effect on the spin lifetime in Supplementary Section 11 shows no clear trend for both spin configurations as the applied current density is smaller than that for self-oscillation.”

Figure R1 | Current-dependent spin lifetime in α -Fe₂O₃.

1. Taking into account the fast decay of the spin current with increasing distance from the interface, the SOT effects are limited to a nanometer-thick layer of Fe₂O₃ near the interface. I find it difficult to understand how one can speak about (modified) magnon

dispersion if the properties of the medium are changed on a spatial scale much smaller than the wavelength of magnons. It is particularly unclear when magnons propagating perpendicular to the interface are considered. This needs to be clarified.

Our reply: We appreciate the reviewer’s feedback. We recognise that the electron spin current cannot propagate over nanometre scales. Therefore, we adopt a magnon-mediated spin current J_m , which can propagate into α -Fe₂O₃. As supported by references [*Nat. Nanotechnol.* **15**, 563 (2020) and *Nature* **561**, 222 (2018)], α -Fe₂O₃ exhibits a magnon conductivity of $\sim 10^5$ S m⁻¹ and a decay length ≥ 250 nm which is longer than the magnon wavelength of ~ 150 nm at room temperature. Using these key parameters, we derived a magnon current profile, including the effective spin mixing conductance between Pt and α -Fe₂O₃ (see Figure R2). The newly adopted J_m reproduces the dispersion relation in our original manuscript.

In the revised manuscript and Supplementary Section 9, we introduce magnon-mediated spin currents for understanding the modified magnon dispersion as follows. “To understand the modified magnon dispersion, we adopt a magnon-mediated spin current J_m , which can propagate into α -Fe₂O₃ (see Method and Supplementary Section 9).”

Figure R2 | Magnon current profile for $J_c = 2 \times 10^7$ A cm⁻² in α -Fe₂O₃.

2. At what depth are magnons detected by the magneto-optical technique? As far as I understand, to achieve efficient Bragg reflection, this depth must be at least several wavelengths of magnons, i.e. several hundred nanometers. At such depths, the effects of SOT become negligible (see the comment above). Thus, it is unclear, how to correctly interpret the results of the measurements. I would expect the detected magnons to have dispersion unaffected by SOT.

Our reply: We thank the reviewer's good comment. The detection depth of the magneto-optical technique is determined by the penetration length of the probe beams,

which is around 300 nm in this work. As discussed in Q1, we adopt a magnon-mediated spin current inside α -Fe₂O₃, and the magnon decay length > 250 nm is longer than that of the magnon wavelength (~ 150 nm). Therefore, Bragg's condition is satisfied.

As the magnon current density decreases away from the α -Fe₂O₃/Pt interface, as shown in Fig. R2, the corresponding spectrum at f_m would experience spectral broadening with a reduced peak amplitude. In the experiment, spectra at $J_c = 0$ and 2×10^7 A cm⁻² are shown in Figure R3, where the magnon amplitude decreased from 17.9 to 15.5, increasing from $J_c = 0$ to 2×10^7 A cm⁻². In the inset of Figure R3, we observe a slight increase of normalised linewidth (df_m / f_m , df_m is the full width at half maximum of the magnon spectra) with increasing J_c . For clarity, we have added discussions on the detection depth of the magneto-optical technique in the Methods and Supplementary Section as follows.

In Methods: “In this case, the magnon detection depth is order of the penetration length of probe beams in α -Fe₂O₃, which is around 300 nm⁵². In this case, the reduced amplitude of SOT away from the α -Fe₂O₃/Pt interface will potentially lead to the broadening of linewidth and reduced amplitude of magnon spectra in the magneto-optical results (Supplementary Section 14).”

In Supplementary Section 14: “As discussed in Section 9, the magnon detection distance of the magneto-optical method is comparable to that of the magnon decay length. In this case, the reduced amplitude of SOT away from the α -Fe₂O₃/Pt interface potentially leads to the broadening of linewidth and reduced amplitude of magnon spectra in the magneto-optical results. To confirm this, the magnon spectra in the frequency domain at $J_c = 0$ and 2×10^7 A cm⁻² are shown in Fig. S3. The magnon amplitude decreases from 17.9 to 15.5 with increasing $J_c = 0$ to 2×10^7 A cm⁻². In the inset of Fig. S3, we observe a slight increase of normalised linewidth (df_m/f_m , df_m is the full width at half maximum of the magnon spectra) with increasing J_c .”

Figure R3 | Magnon spectra at $J_c = 0$ and 2×10^7 A cm⁻² in α -Fe₂O₃. The inset is the corresponding normalised linewidth.

3. From a comparison of the data shown in Fig. 2c and the data in Fig. 3c ($J_c=0$), it can be seen that the deposition of Pt leads to an increase in the frequency of magnons from about 250 to 270 GHz. In other words, the presence of Pt strongly modifies the properties of the system. Therefore, it is inaccurate to refer to the data obtained for bare Fe₂O₃ when discussing SOT effects in Fe₂O₃/Pt. In particular, in Fe₂O₃/Pt, the temperature dependence can differ from that shown in Fig. 2c. The temperature dependence must be measured for the Fe₂O₃/Pt heterostructure. This should not be a problem for the magneto-optical technique.

Our reply: We appreciate the reviewer’s helpful comment. We combine the data points in both Fig. 2c and Fig. 3c ($J_c = 0$), and the magnon frequency does not increase after depositing Pt as shown in Figure R4. The slight frequency gap is from the different measurement temperatures (290 K vs. 300 K) between the two datasets.

Figure R4 | comparison of the magnon frequency dataset between Fig. 2c and Fig. 3c in the main text. The magnon frequency is not increased after the deposition of Pt.

To further check the modified effect of Pt on the α -Fe₂O₃ magnon, the temperature-dependent magnon frequency for both bare α -Fe₂O₃ and α -Fe₂O₃/Pt are measured from 50 K to 300 K as shown in Figure R5. No effect of Pt on α -Fe₂O₃ magnon is observed at the whole temperature range.

Figure R5 | Temperature-dependent magnon frequency f_m of bare Fe₂O₃ and Fe₂O₃/Pt (5 nm). The inset images show the measurement spots. No effect of Pt is observed at the whole temperature range.

4. How can we be sure that the observed effects are due to SOT in Fe₂O₃ and not due, for example, to modification of proximity magnetism in Pt?

Our reply: Thanks for your helpful comment. We confirm that the observed f_m tuning effects are due to SOT for several reasons. First, the nonlinearly decreased trend of f_m with J_c matches well with both previous calculation works [*Phys. Rev. Lett.* **116**, 207603 (2016); *Appl. Phys. Lett.* **117**, 222411 (2020)] and our spin-wave model. Especially, the observed f_m effects reveal distinct dependences on the spin configurations as shown in Fig. 3c, which matches well with the SOT behaviour that the tuning becomes prominent at $\sigma \perp \mathbf{l}$ ($\sigma // \mathbf{m}$) in Fe₂O₃. Besides, we have conducted measurements to rule out other spurious effects, including current-induced heating (Fig. 2c and Extended Data Fig. 8), strain (Fig. 4a) and magnetic field effects (Extended Data Fig. 10). For instance, we measured the current-induced effect on f_m using α -Fe₂O₃/Cu (5 nm) control device. As shown in Fig. 4a, f_m is not tuned by J_c at both spin configurations, which strongly suggests that SOT is essential for the observed f_m tuning rather than the current-induced strain effect. In addition, we model the current-induced heating effect. The temperature rise may slightly increase f_m at $T > T_M$ (Fig. 2c), which contrasts with the observed decrease trend of f_m with J_c in Fig. 3c (Supplementary Section 10). Moreover, the result in Figure R5 shows no modification of proximity magnetism in Pt. Therefore, we conclude that the SOT dominates the magnon frequency tuning in this work.

5. Why are measurements of the main effect (Fig. 3c) limited by $J_c=2 \times 10^7$ A/cm²? According to the authors' calculations, the increase of the temperature is rather small at these densities. Thus, the measurement range can be easily extended to much higher currents. It should be possible to go up to 10^8 , as was done in other works for similar

dimensions of Pt electrodes. If the authors are correct in their interpretation, then the frequency should start to increase at a certain current due to the effects of temperature. These data must be added to make the interpretation convincing.

Our reply: Thanks to the reviewer's insightful comment on further increasing the current density due to the thermal effect. We acknowledge that an increase in magnon frequency with further augmentation of J_c may be possible due to thermal effects. Nevertheless, thermal disturbance can also disturb the magneto-optical signal in actual experiments, rendering it difficult to observe. Figure R6 demonstrates the magneto-optical measurement results of α -Fe₂O₃/Pt at a higher J_c up to 3×10^7 A cm⁻². As shown in the inset of Figure R6b, the magnon amplitude starts to drop with increasing J_c at $J_c > 1.5 \times 10^7$ A cm⁻². To estimate the current-induced temperature rise, we simulate the relationship between the temperature on the α -Fe₂O₃/Pt interface and J_c , which is shown in Figure R7. The rise in temperature is proportional to the square of J_c . Therefore, the temperature rise at $J_c = 2 \times 10^7$ A cm⁻² is only ~ 6 K, but it increases dramatically to over 100 K when J_c increases to 1×10^8 A cm⁻². This reduced magnon amplitude due to thermal effect causes incoherency of magnon and therefore hinders our measurement at higher J_c values. Therefore, we were limited to analyze the data with $J_c \leq 2 \times 10^7$ A cm⁻². In addition, we conduct the magneto-optical measurement using the laser spot positioned next to the Pt electrode in Figure R8 to confirm the pure thermal effect. The increase in magnon frequency due to the thermal effect is observed.

We have added the discussions in Methods and Supplementary Section 15 as follows; “The limitation of the measurement scheme is that the thermal effect disturbs the magnon signal at higher $J_c > 2 \times 10^7$ A cm⁻², which prevents from conducting any meaningful analysis (details are shown in Supplementary Section 15). Therefore, we restrict the data analysis to $J_c < 2 \times 10^7$ A cm⁻² in this work.”.

Figure R6 | Current dependent magnon amplitude in Fe₂O₃. **a** Time-resolved polarisation rotation results under various J_c and **b** the corresponding magnon spectra. The magnon amplitude reduces with J_c especially when $J_c > 1.5 \times 10^7$ A cm⁻². The inset indicates that peak magnon amplitude starts to decrease at $J_c > 1.5 \times 10^7$ A cm⁻².

Figure R7 | Simulated current density versus temperature on the Pt/ α -Fe₂O₃ interface. The temperature rise is proportional to the square of the current density.

Figure R8 | Current dependent magnon frequency on the surface of bare α -Fe₂O₃ next to the Pt electrode. The inset indicates the spot position on the left or right of the Pt electrode.

6. An optional but very interesting extension would be a measurement similar to that shown in Fig. 3c, with the laser spot placed on the surface of bare Fe₂O₃ next to the Pt electrode. These data could very convincingly demonstrate that the effects exist only underneath the Pt strip.

Our reply: Thanks for this helpful comment. Figure R8 demonstrates that the magnon frequency results on the surface of bare α -Fe₂O₃ next to the Pt channel (laser spots are placed on the either side of the Pt channel). The monotonically increased magnon frequency with increasing the current contrasts with the observed frequency reduction in Fig. 3c in the main text. This result supports that the observed frequency reduction only exists underneath the Pt strip due to SOT.

Reviewer #3:

Yang et al. reported spin-orbit torque manipulation of sub-terahertz magnons in antiferromagnet. Authors detect sub-terahertz spin precession in alpha-Fe₂O₃ by time-resolved pump-probe technique with injecting current in the heterostructure. They observed first time that spin precession frequency of antiferromagnetic alpha-Fe₂O₃ can be modulated by the spin-orbit torque in alpha-Fe₂O₃/Pt heterostructure. The polarity of precession frequency change is opposite with the trend of Joule heating; therefore, the change is certainly caused by the spin-orbit torque. I feel that this study contains many novelties therefore this study is valuable to be published in Nature Communications. But I have following questions and comments, which must be addressed before acceptance.

Our reply: We thank the reviewer for finding our work novel. In the reply below, we addressed the reviewer's comments and improved the manuscript accordingly.

1. Authors used second-ordered magneto-optical effect such as Cotton-Mouton effect to detect antiferromagnetic resonance. Usually, this effect is birefringence effect and cannot be detected by measuring rotation of electric field. Actually, the previous study (Ref. 29) used quarter-wave plate to detect ellipticity of probe beam, however, authors use half-wave plate to detect rotation of probe beam. It should be mentioned that why they measured rotation instead of ellipticity in the revised manuscript.

Our reply: We sincerely appreciate your insightful comment regarding the foundational aspects of our magneto-optical measurement technique. It is accurate to assert that the magnetic linear birefringence cannot be effectively demodulated by a half-wave plate (HWP). However, in cases where magnetic linear birefringence and dichroism simultaneously manifest, the polarisation and ellipticity of the probe beam become subject to tuning. In such circumstances, the oscillating signal emanating from magnons can be detected through the utilisation of either a HWP or by strategically manipulating the optical axis of a Wollaston prism [Coey et al., Chapter 5, "Magnetism and Magnetic Materials"; *Phys. Rev. B* **95**, 174407 (2017); *Nat. Phys.* **17**, 1001–1006 (2021)].

We conducted a control experiment employing a quarter wave plate (QWP) as the analyser. As shown in Figure R9, the oscillating signals stemming from AFM magnons can be demodulated through either the HWP or QWP. It is noteworthy to observe that the results obtained via the QWP also encompass the presence of an incoherent baseline signal stemming from spin and electron relaxations. In a stark contrast, the HWP exhibits a clean baseline, particularly in the lower frequency regime below 100 GHz, as shown in Figure R9b. Therefore, we have adopted the HWP for detecting magnon signals in this work. We have added descriptions in Method as follows:

"...Reflectivity and the polarisation rotation of the probe beams were measured by

using a Wollaston prism and a balanced detector combination (Extended Data Fig. 1). To be noted, the pure magnetic linear birefringence cannot be demodulated by using an HWP. However, in cases where both magnetic linear birefringence and dichroism simultaneously manifest, both the polarisation and ellipticity of the probe beam become subject to tuning. In such circumstances, the oscillating signal emanating from magnons can be detected through the utilisation of either a HWP or by strategically manipulating the optical axis of a Wollaston prism.”

Figure R9 | Transient magneto-optical measurement of bare α -Fe₂O₃ by using the HWP and QWP as the analyser, respectively. **a The oscillating signals of coherent AFM magnons are demodulated with both waveplates in the time domain. A HWP is better for filtering the signals from incoherent spin and electron relaxations. **b** The corresponding FFT spectra of **a**. Both acoustic phonon and magnon modes are distinguished with both waveplates, but the baseline measured by a HWP is cleaner at a lower frequency range below 100 GHz.**

2. Authors mentioned in Supplementary Information section 3 that q-FM magnon mode reverses with the direction of magnetic field, but they only compare with and without magnetic field in Extended Data Fig. 5.

Our reply: Thanks for the reviewer’s valuable comment. In response, we have included a comparison between the magnon signals in the presence of 300 mT positive (H^+) and negative (H^-) magnetic fields. Figure R10 presents the amplitude of the sum ($H^+ + H^-$) and difference ($H^+ - H^-$) of magnon signals. If the phase of the q-FM magnons mode reverses with the direction of the magnetic field, its peak should be highlighted in the difference ($H^+ - H^-$) result. This is confirmed in Figure R10b. We have added the above explanations in the main text and updated Extended Data Fig. 5.

Figure R10 | Symmetry analysis of q-FM magnon in α -Fe₂O₃ under the in-plane external magnetic field. c, The sum ($H^+ + H^-$) and difference ($H^+ - H^-$) of the oscillating signals. d, Corresponding FFT spectra. The peak of the q-FM magnon should be observed in the difference ($H^+ - H^-$) result if the phase of the q-FM magnon mode reverses with the opposite direction of the magnetic field.

3. In Fig. 3c, it looks that experimental data has large scattering, i.e., there is deviation between experimental symbols and solid curves. Where does this scattering come from? In Fig. 4b and Extended Data Table 1, authors defined tuning efficiency of frequency and compare with other reports. Is it better to show error bar in the tuning efficiency because there is scattering in the experimental data?

Our reply: We appreciate the reviewer's comment. This scattering comes from the limited signal-to-noise ratio of magneto-optical measurement as the amplitude of light is reduced by the covered Pt layer, and the duty cycle is reduced to minimise the thermal effect. We agree that it is better to show the error bar in the tuning efficiency accordingly. The revised Fig. 4b and Extended Data Table 1 are shown below:

Revised Figure 4b

Revised Extended Data Table 1.

Materials	f_m (GHz)	η (/ 10^7 A cm $^{-2}$)	Tunability	Measurement condition	Driving force	Ref.
α -Fe $_2$ O $_3$	272 ($k=4.08 \times 10^5$ cm $^{-1}$)	-2.6 ± 0.4 % ($k=4 \times 10^5$ cm $^{-1}$)	-14 ± 2.0 GHz/ 2×10^7 A cm $^{-2}$ ($k=4.08 \times 10^5$ cm $^{-1}$)	Room temp.	SOT ($\sigma // m$)	This work
	170 ($k=0$)	-6.8 ± 1.0 % ($k=0$)	-23 ± 3.1 GHz/ 2×10^7 A cm $^{-2}$ ($k=0$)			

4. In Fig. 4a, authors show the results of control samples, that is alpha-Fe $_2$ O $_3$ /Cu. But the range of vertical axis is different with the case for alpha-Fe $_2$ O $_3$ /Pt sample (Fig. 3c). It is better to compare with same range of vertical axis.

Our reply: Thank you very much for pointing this out. We have set the same range for both Fig. 3c and Fig. 4a for comparison. The revised Fig. 4a is shown below.

Revised Figure 4a

5. As shown mainly in Supplementary Information, the authors explain the antiferromagnetic resonance by the free energy including homogeneous and inhomogeneous exchange energy, uniaxial magnetic anisotropy energy, and DM interaction energy, but I cannot find the parameters (values) they used exactly in the manuscript. This must be described in the revised version.

Our reply: We appreciate the reviewer's constructive suggestion. In the revised manuscript, the fitting parameters are summarised in the Methods section “Atomistic spin simulations for α -Fe $_2$ O $_3$ magnon”. We have also added the cross-citation in Supplementary information in the modified manuscript; “The exact values of these parameters are given in the Methods section.”.

6. Authors derived equations for antiferromagnetic resonance frequency in Supplementary information but the details of parameters are not written in the text. For

example, resonance frequency is expressed in terms of effective magnetic field due to exchange interaction or DM interaction, but their correspondence with energy (i.e., a , A , D) are not described. This must be added for readers.

Our reply: We appreciate the review's suggestion. We have added the descriptions in Supplementary Section 1 as follows for readership.

"The a and A are the homogeneous and inhomogeneous exchange constant, respectively; the parameters are defined as $A = \Delta_a^2 \hbar \omega_E$, $a = 2 \hbar \omega_E$, and $\mathbf{D} = \hbar \omega_D \mathbf{e}_y$ and Δ_a is set to the length between neighbours along the z -axis. K_z and K_y are the uniaxial anisotropy along the z and y direction, respectively."

7. As shown in Fig. 3c, d, the precession frequency is quadratically decreased with increasing current for both cases that spin is parallel and perpendicular to Neel vectors. This trend is explained by the theory of antiferromagnetic magnon including spin-orbit torque term as shown in Eq. (S12a) and Eq. (S13a), where the spin-orbit torque modifies $k = 0$ magnon frequency as well as equilibrium magnetisation angle. By analysing the results, authors can obtain the parameters of spin-orbit torque effective magnetic field or spin-Hall angle in Pt, however, they did not describe exact values in the text. This must be addressed in the revised version.

Our reply: We thank the reviewer's insightful comment. In the revised version, we added how to obtain modified magnon frequency using parameters in the revised version. In Section 7, we add one example to obtain magnon frequency at $f_m(0.0408) = 258.795$ GHz, where the effective SOT field is calculated using magnetic parameters in Methods. We have added below discussions in the main text:

"...For example, at $J_c = 2 \times 10^7$ A cm⁻², $f_m(0.0408) = 258.8$ GHz where $H_{\text{SOT}} = 2\pi J_c \zeta / \gamma = 404.7$ Oe, $\Phi_0 = 1.2$ rads (see Methods for parameters)..."

8. Regarding spin-orbit torque strength parameter zeta, this strength is inversely proportional to thickness of antiferromagnetic layer, but the thickness of the antiferromagnetic layer is 1 nm as described in Methods section, quite thin, so the spin-orbit torque strength becomes quite weak. In the Atomistic spin simulation section, authors state that the effective thickness of antiferromagnetic layer used was 4 nm. It must be stated why they chose this value.

Our reply: We appreciate the reviewer's feedback. We recognise that the electron spin current cannot propagate over nanometre scales. Therefore, we adopt a magnon-mediated spin current J_m , which can propagate into α -Fe₂O₃. As supported by references [*Nat. Nanotechnol.* **15**, 563 (2020) and *Nature* **561**, 222 (2018)], α -Fe₂O₃ exhibits a magnon conductivity of $\sim 10^5$ S m⁻¹ and a decay length of ~ 250 nm which is longer than

the magnon wavelength of ~ 150 nm at room temperature. We derived a magnon current profile using these key parameters, including the effective spin mixing conductance between Pt and α -Fe₂O₃ (Figure R11). The newly adopted J_m reproduces the dispersion relation in our original manuscript.

In the revised manuscript and Supplementary Section 9, we introduce magnon-mediated spin currents for understanding the modified magnon dispersion as follows. *“To understand the modified magnon dispersion, we adopt a magnon-mediated spin current J_m , which can propagate into α -Fe₂O₃ (see Method and Supplementary Section 9).”*

Figure R11 | Magnon current profile for $J_c = 2 \times 10^7$ A cm $^{-2}$ in α -Fe₂O₃.

Reviewers' Comments:

Reviewer #1:

Remarks to the Author:

The authors have addressed the points raised by the referees, now it can be accepted for publication in Nature Communications.

Reviewer #2:

Remarks to the Author:

I thank the authors for their detailed response to my comments. Their response and changes in the manuscript address some of the issues mentioned in my first report. However, there are still significant inconsistencies in the experimental results and their interpretation that need to be resolved before the manuscript can be published.

1. The authors recognized the problem with the short penetration length of the electron spin current. To solve this problem, they now use in their model a magnon-mediated spin current, which has a much longer penetration length. However, the efficiency of conversion of electron spin current into magnon-mediated spin current is quite low. Therefore, one would expect all effects to become much weaker. In their model, the authors assume that the chemical potential is continuous at the interface. This is obviously incorrect (see, e.g., Fig. 2 and the corresponding discussions in Phys. Rev. B 94, 014412 (2016)). In order for the model to be realistic, the finite spin conductance of the interface must be taken into account.

2. The authors introduce a parameter called "space-averaged spin-to-magnon conversion ratio $\gamma_m = 0.011$ (0.013)", but do not explain anywhere how it was calculated and whether the obtained value is reasonable.

3. The authors say that "the thermal effect disturbs the magnon signal at higher $J_c > 2 \times 10^7$ A cm⁻², which prevents from conducting any meaningful analysis." However, when measuring next to the Pt electrode, where the temperature should be almost the same, they are able to obtain a magnon signal at current densities up to 5×10^7 . This indicates that the temperature IS NOT a factor leading to suppression of the magnon signal. Additionally, according to the author's simulations, at 3×10^7 A cm⁻² the temperature increase does not exceed 20 K. It is very unlikely that such a small increase affects the magnon signal so strongly. Therefore, it is more reasonable to assume that this is the effect of SOT. If it were a thermal effect, it would also be clearly visible when measuring near the electrode.

4. As seen from Fig. S4, increasing the current density above 2×10^7 A cm⁻² does not significantly reduce the magnon signal itself. Instead, it leads to a decrease in the magnon lifetime. This indicates that SOT does not decrease, but rather strongly increases the damping. This is an important unexpected finding, which must be discussed and explained in the manuscript. It is also important to add these data to Fig. S2.

Reviewer #3:

Remarks to the Author:

Authors modified points which I mentioned in the previously review.

Now I feel that their experiment and analysis are correct.

I think that this work contains many novelties.

Therefore, I recommend the manuscript to be published in Nature Communications.

Reviewer #2:

I thank the authors for their detailed response to my comments. Their response and changes in the manuscript address some of the issues mentioned in my first report. However, there are still significant inconsistencies in the experimental results and their interpretation that need to be resolved before the manuscript can be published.

Our reply: We sincerely express our gratitude to the reviewer for his/her further invaluable comments. We have addressed each comment in a point-by-point manner.

1. The authors recognized the problem with the short penetration length of the electron spin current. To solve this problem, they now use in their model a magnon-mediated spin current, which has a much longer penetration length. However, the efficiency of conversion of electron spin current into magnon-mediated spin current is quite low. Therefore, one would expect all effects to become much weaker. In their model, the authors assume that the chemical potential is continuous at the interface. This is obviously incorrect (see, e.g., Fig. 2 and the corresponding discussions in Phys. Rev. B 94, 014412 (2016)). In order for the model to be realistic, the finite spin conductance of the interface must be taken into account.

Our reply: We sincerely thank the reviewer's comments. As per the reviewer's comment, the magnon current density is low. However, α -Fe₂O₃ has a higher exchange field ($H_E \sim 930$ T) and lower anisotropy field ($H_A \sim 168$ Oe) compared to other antiferromagnets (e.g. YFeO₃ has a lower exchange field of 635 T and higher anisotropy field of ~ 7000 Oe, *Nat. Commun.* **13**, 6140 (2022)). Therefore, even a low magnon current density is expected to modify the magnon frequency in α -Fe₂O₃.

The chemical potential must be discontinuous at the interface. By introducing effective spin conductance, we obtained discontinuous magnon chemical potential that is shifted downward compared to spin chemical potential on the Pt-side. Here, the interface spin current is defined as $J_s^{\text{int}} = g_s(u_s^{\text{int}} - u_m^{\text{int}})$ where g_s is the effective spin conductance at the interface. We added Fig. S1a and S1b for magnon spin current and chemical potential in Supplementary Note 9.

Figure S1| Magnon chemical potential (a) and current density (b) for $J_c = 2 \times 10^7$ A cm⁻² in α -Fe₂O₃. Distance (d) is the depth along z direction from the α -Fe₂O₃/Pt interface.

2. The authors introduce a parameter called “space-averaged spin-to-magnon conversion ratio $\gamma_m = 0.011$ (0.013)”, but do not explain anywhere how it was calculated and whether the obtained value is reasonable.

Our reply: Thanks for this helpful comment. Rather than using a new parameter (spin-to-magnon conversion ratio), we now directly show the profiles of magnon current density and chemical potential, calculated from the effective spin conductance at the interface and spin current injected from Pt.

3. The authors say that “the thermal effect disturbs the magnon signal at higher $J_c > 2 \times 10^7$ A cm⁻², which prevents from conducting any meaningful analysis.” However, when measuring next to the Pt electrode, where the temperature should be almost the same, they are able to obtain a magnon signal at current densities up to 5×10^7 . This indicates that the temperature IS NOT a factor leading to suppression of the magnon signal. Additionally, according to the author’s simulations, at 3×10^7 A cm⁻² the temperature increase does not exceed 20 K. It is very unlikely that such a small increase affects the magnon signal so strongly. Therefore, it is more reasonable to assume that this is the effect of SOT. If it were a thermal effect, it would also be clearly visible when measuring near the electrode.

Our reply: We appreciate the reviewer’s comment. To quantitatively compare the temperature rise beneath and near the Pt electrode, we extracted the simulated temperature distribution of α -Fe₂O₃ in Figure R1 with $J_c = 2 \times 10^7$ A cm⁻². The temperature rise $\Delta T = T - T_0 \sim 3$ K near the electrode ($Y = \pm 4$ μ m) is less than half of the temperature rise ~ 7 K beneath the Pt electrode ($Y = 0$ μ m). This is due to the poor thermal conductivity of insulating α -Fe₂O₃ (12.7 W/(m·K)) and the small duty cycle for the applied current. As we increase the current density, the temperature difference between two locations would be even larger.

Figure R1 | Simulation for current-induced heating effect. **a**, Simulation schematic of $\alpha\text{-Fe}_2\text{O}_3/\text{Pt}$ (5nm) devices with $J_c = 2 \times 10^7 \text{ A cm}^{-2}$. The temperature profile of $\alpha\text{-Fe}_2\text{O}_3$ along the white dash line is shown in **b**. The yellow region in **b** indicates the 4 μm Pt channel. $T_0 = 293.15 \text{ K}$ is the base temperature.

To further confirm the thermal effect on magnon signals, the temperature-dependent magnon amplitude are measured between 100 and 300 K without applying the current in Figure R2. The magnon amplitude reaches its maximum at around $T_M \sim 260 \text{ K}$, and then monotonically drops with further increasing temperature. This result provides direct evidence that the reduction in magnon amplitude is caused by the thermal effect.

Figure R2 | Temperature-dependent magnon amplitude in $\alpha\text{-Fe}_2\text{O}_3/\text{Pt}$ (5nm).

4. As seen from Fig. S4, increasing the current density above $2 \times 10^7 \text{ A cm}^{-2}$ does not significantly reduce the magnon signal itself. Instead, it leads to a decrease in the magnon lifetime. This indicates that SOT does not decrease, but rather strongly increases the damping. This is an important unexpected finding, which must be discussed and explained in the manuscript. It is also important to add these data to Fig. S2.

Our reply: We thank the reviewer's comment. To confirm whether the magnon amplitude or magnon lifetime is altered, we extracted the magnon amplitude and linewidth (which is inversely proportional to the magnon lifetime) by performing a

Gaussian fitting on Figure S4b. The results are presented in Figure R3. The magnon amplitude decreases with increasing the current density, at $J_c > 1.5 \times 10^7 \text{ A cm}^{-2}$. Comparatively, no clear trend is observed for the magnon linewidth.

Figure R3 | Current-dependent magnon amplitude and magnon lifetime in $\alpha\text{-Fe}_2\text{O}_3$.

Reviewers' Comments:

Reviewer #2:

Remarks to the Author:

The authors have adequately addressed my concerns. I recommend publication of the manuscript in its current form.